# The downward spiralling nature of the North Atlantic Subtropical Gyre

Sara Berglund [1]✉, Kristofer Döös [1], Sjoerd Groeskamp [2] & Trevor J. McDougall[3]

The Atlantic Meridional Overturning Circulation (AMOC) regulates the heat distribution and climate of Earth. Here we identify a new feature of the circulation within the North Atlantic Subtropical Gyre that is associated with the northward flowing component of the AMOC. We find that 70% of the water that flows northwards as part of the AMOC circulates the Gyre at least once before it can continue northwards. These circuits are needed to achieve an increase of density and depth through a combination of air-sea interaction and interior mixing processes, before water can escape the latitudes of the Gyre and join the northern upper branch of the AMOC. This points towards an important role of the Gyre circulation in determining the strength and variability of the AMOC and the northward heat transport. Understanding this newly identified role of the North Atlantic Subtropical Gyre is needed to properly represent future changes of the AMOC.

[1] Department of Meteorology, Stockholm University, Stockholm, Sweden. [2] NIOZ Royal Netherlands Institute for Sea Research, Den Burg, Texel, The Netherlands. [3] School of Mathematics and Statistics, University of New South Wales, Sydney, NSW, Australia. ✉email: sara.berglund@misu.su.se

The Atlantic Meridional Overturning Circulation (AMOC) has most likely slowed down during the 20th century[1–4] and is projected to further slow down in the decades to come[5,6]. Since the AMOC influences both local and global climate, understanding the observed changes has become the topic of many studies[1–4]. The Gulf Stream is part of the northward flowing component of the AMOC, as well as the North Atlantic Subtropical Gyre (hereafter referred to as the "Gyre")[7]. Some of the Gulf Stream water is captured in the Gyre[8], where it becomes colder and fresher[9]. The transition between the Subtropical Gyre and the Subpolar Gyre is suggested to occur at depth rather than at the surface[10–14], and this connection at depth could be due to re-circulation in the Subtropical Gyre[11]. Further, in the warm flank of the Gulf Stream, the North Atlantic Subtropical Mode Water (NASTMW) is formed[11,15–18], a vertically homogeneous water mass, most often found over a relativity large geographical area[19]. This water mass is also suggested to be the link between the Subtropical Gyre and Subpolar Gyre at depth[11]. Clearly, the Gyre plays a significant role in the AMOC, but to what extent and how is still unclear.

Northward heat transport by the AMOC[7] is considered to be the reason for the relatively mild climate in Northern Europe[5]. This Atlantic heat transport is strongly influenced by the wind-driven ocean circulation[5,20]. Meanwhile, the water masses in the Gyre are projected to become less dense in a warmer climate, due to a combination of raised temperature and increased salinity[5]. Together, changes in wind patterns and in temperature and salinity of the water masses, may significantly impact the Gyre circulation, and therewith the AMOC and its northward heat transport.

In the present study, the use of Lagrangian trajectories[21,22] confirms qualitatively and quantitatively the important role of the Gyre for the AMOC. Trajectories are simulated using data from an Earth System Model with two different type of resolutions (1° and 1/4°). More details on how this is done is provided in the Method section. We find that 70% of the water that flows northwards first performs between 1 and 14 loops in the Gyre, before it has become sufficiently dense to continue northwards. During these loops air-sea interaction and mixing cause a net densification, allowing for the water to spiral down to greater depths than those where it entered the Gyre. Our results suggest that changes in Gyre dynamics may lead to changes in the AMOC's future pathways and strengths. A connection like this has rarely been made, illustrating an underrepresented role of the Gyre in the discussion about the AMOC and its role in future climate.

## Results

**The role of the Gyre**. The spiralling nature of the Gyre is computed as a mean trajectory from hundreds of thousands of simulated Lagrangian trajectories (Fig. 1, for a detailed description see methods section). This can be interpreted as the mean pathway of the circulation in the Gyre. About 3% of the water that flows northwards as part of the AMOC circulates around the Gyre once, 20% undertakes two circuits, while 47% circulates around the Gyre three or more times before heading north (Table 1). A water parcel caught in the Gyre becomes colder, slightly fresher and denser with each circuit (Fig. 1). Consequently it ends up deeper in the water column where the higher densities are found (Fig. 1a, b). As the parcel orbits at increasingly deeper levels, we find a decrease in the influence of air-sea interaction and an increase in the relative importance of ocean mixing (discussed further in coming sections). Figure 1 describes a mean pathway of all trajectories circulating the Gyre seven or more times. The geographical distribution of parcels are seen in

Fig. 1d. The mean trajectory is, however, based on the volume transport circulating the gyre, meaning that most trajectories will follow a similar path as that illustrated in Fig. 1.

Water enters the Gulf Stream with a density ranging between 24 and 25 kgm$^{-3}$ in the upper 100 metres of the ocean (Fig. 2 at 5°N). As the water exits the Gulf Stream in the north, it does so with a density larger than 27 kgm$^{-3}$, spread at depths between 250 and 1000 metres depth (Fig. 2 at 55°N). In other words, water that comes into the Gulf Stream from the south is lighter than the same water that later moves out in the north (Fig. 2b). Although 30% of the water flows out directly (while the density changes along the way), the other 70% of the water first has to gain density by completing one or more Gyre spirals. The water that spirals the Gyre is also somewhat deeper and denser than that flowing directly northward (Fig. 2).

**The spiralling Gyre**. The use of many single Lagrangian trajectories makes it possible to compute the mean trajectory in Fig. 1a–c, e (method section). The mean trajectory (Fig. 1) is used in the coming section to examine the temperature, salinity, density and depth changes experienced during each circuit of the Gyre.

Parcels circulating in the Gyre for the first time, do so close to the surface (red line Fig. 1a–c). These parcels experience strong cooling and become denser, as they flow east as part of the Gyre (Fig. 1d, e red–green colour). When parcels have moved to the east Atlantic, cooling continues while salinity increases and the parcels continue to become denser (green in Fig. 1d, e). In the southeastern parts of the Gyre, the parcels temperature and salinity both increase (dark green in Fig. 1d, e), and does so while the density is unchanged. As they reach the most southern parts of the Gyre, their temperature continue to increase but salinity stays rather constant, which results in a density decrease (blue in Fig. 1d, e). Before the parcels reach the Gulf Stream again, they become slightly lighter, as a result of freshening in the southwestern parts of the Gyre (purple in Fig. 1d, e). When the parcels finish their loop and finally return to the Gulf Stream they have become colder, more saline and denser, and reside at greater depths than when they entered the Gyre (Fig. 1). That is, during a circuit of the Gyre, some initial increase of density is subsequently compensated in some regions where the water actually becomes lighter, but the net result for each circuit is an increase of density. This first circuit is an outlier compared to the rest of the circuits (red line Fig. 1), as it stretches over a large density range and is the only circuit where the parcel's salinity is increased. Consequently, this leads to the most significant depth change over one circuit, compared to the following circuits (Table 1).

About 67% (Table 1) of the parcels will undergo at least one more circuit to become denser (blue line Fig. 1c). This second circuit is different from the first (Fig. 1a and Table 1), but similar to the third, forth and fifth circuits. For each of these circuits the parcels experience a similar type of forcing that we describe below (blue line and beyond in Fig. 1a–c). As a parcel flows along the Gulf Stream and the northernmost parts of the Gyre it is both cooled and freshened, while it's density increases (red–light green in Fig. 1d, e). In the eastern parts of the Gyre, the parcel cools and becomes more saline, and thus continues to increase in density (green in Fig. 1d, e). Thereafter, as the parcel turns westward, both temperature and salinity increases, while the density is kept rather constant (dark green in Fig. 1d, e). In the western parts of the Gyre, the parcel starts to freshen again, while the temperature continues to increase, which results in a decrease in density (blue–purple in Fig. 1d, e). Finally, the parcel returns to the Gulf Stream again. For each of these circuits the water becomes slightly colder, fresher and denser, and is consequently located deeper

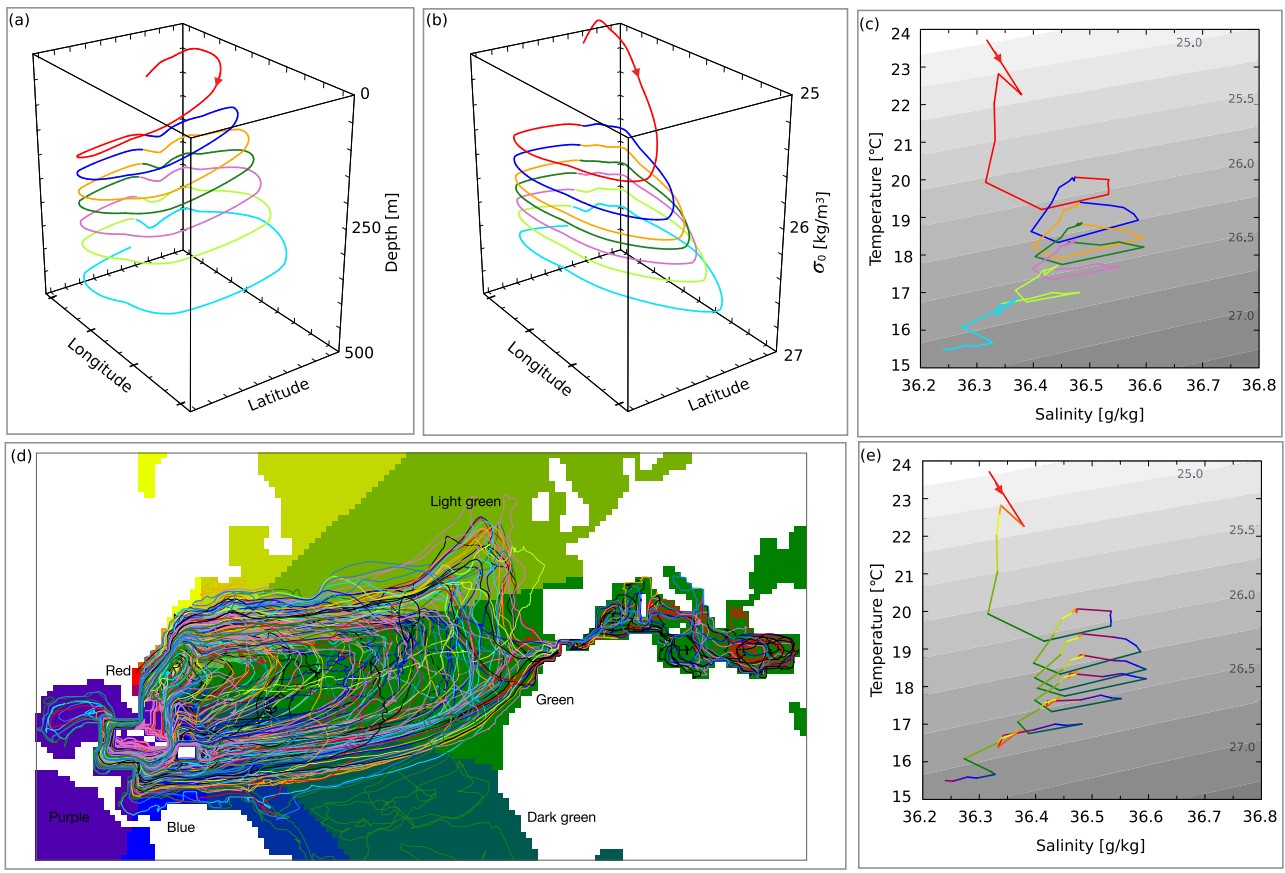

**Fig. 1 The Downward Spiral.** A mean trajectory computed from all trajectories that goes through seven or more circuits in the Gyre. Each circuit is coloured with one colour except for (**e**), which is coloured following the colour–clock in (**d**). **a** 3D illustration of the average trajectory in depth, latitude and longitude coordinates. **b** 3D illustration of the average trajectory in $\sigma_0$, latitude and longitude coordinates. **c** The average trajectory in temperature-salinity space. Grey contours are $\sigma_0$-surfaces. **d** A set of trajectories simulated in the present study that circuits the Gyre. Only their circulation in the Gyre is plotted. The colour wheel is used to compute the average trajectory in (**a**, **b**, **c**, **e**). The names of the colours are included as they are discussed in the text. **e** The average trajectory coloured after location following the colour-clock superimposed in (**d**).

**Table 1 The total volume transport (percentage of the total volume transport reaching north) (1 Sv = $10^6$ m$^3$ s$^{-1}$), average heat change, average salt change, average density increase, average depth increase and average circuit time for all waters in each circuit of the Gyre.**

| Circuit | Volume transport [Sv] (%) | Heat [TW] | Salt [$10^6$ kg/s] | $\Delta\sigma_0$ [kg/m$^3$] | $\Delta z$ [m] | Time [years] |
|---|---|---|---|---|---|---|
| 1 | 7.7 (70%) | −134 | 0.15 | 1.05 | 182 | 22 |
| 2 | 7.4 (67%) | −41 | −0.39 | 0.18 | 48 | 38 |
| 3 | 5.2 (47%) | −24 | −0.34 | 0.12 | 35 | 46 |
| 4 | 3.7 (34%) | −16 | −0.26 | 0.08 | 29 | 57 |
| 5 | 2.6 (24%) | −10 | −0.18 | 0.05 | 22 | 68 |
| 6 | 1.8 (16%) | −7 | −0.12 | 0.03 | 15 | 77 |
| 7 | 1.3 (12%) | −4 | −0.08 | 0.01 | 8 | 86 |

The heat (salt) change describes how much heat (salt) all parcels have lost/gained for one specific circuit. The depth and density increase shows how much the depth and density increase by average for all waters in each circuit of the Gyre. The average time of each circuit shows in general how long it takes for waters to follow that specific circuit.

and deeper (orange line and beyond Fig. 1). However, the difference in depth between start and end of each subsequent circuit, becomes smaller and smaller (Fig. 1e and Table 1). For each circuit, the strong densification in the north and eastern parts of the Gyre are compensated by a decrease in density in the southernmost region of the Gyre (blue–purple in Fig. 1). The big difference in behaviour between the first and subsequent circuits, and especially the reduced changes in salinity and temperature, reinforce the idea that deeper parcels are less influenced by

surface forcing and more by interior ocean mixing processes. This inference needs to be investigated in detail.

In the last two circuits (light green and cyan, Fig. 1a–c), the parcel undergo a larger freshening than for previous circuits. This is a combined effect of more freshening in regions where previous circuits also freshened, but also due to a smaller salinity increase in the eastern parts of the Gyre. These circuits also experience larger net cooling, because of less warming in the southern parts of the Gyre circuit, compared to previous circuits.

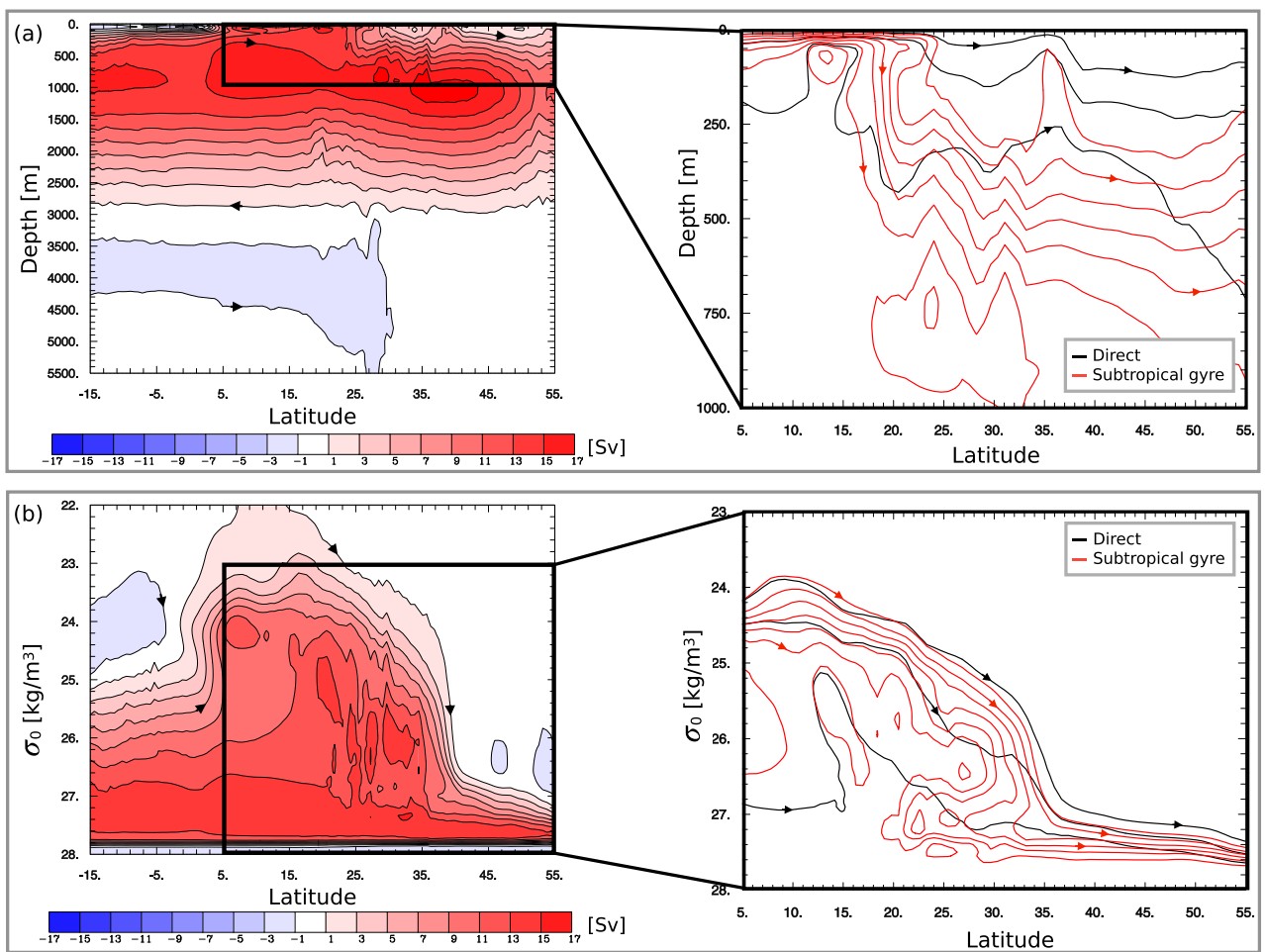

**Fig. 2 The Atlantic Meridional Overturning circulation. a** Left: The Eulerian Atlantic Meridional Overturning stream function computed for the years 1850–2014. The rectangle indicates the area shown in the figure to the right. The contour interval is 2 Sv. Right: The Lagrangian Meridional Overturning stream function for waters that move directly northwards without spiralling in the Gyre (black contours) and for those waters that spiral in the Gyre at least once (red contours). The contour interval is 1 Sv. **b** Left: The Eulerian Atlantic Meridional Overturning stream function in $\sigma_0$-latitude coordinates computed for the same period as in (**a**). The black rectangle indicates the region shown in the right panel. The contour interval is 2 Sv. Right: The Lagrangian Meridional Overturning stream function in $\sigma_0$-latitude coordinates for waters that move directly northwards without spiralling in the Gyre (black contours) and for those waters that spiral in the Gyre at least once (red contours). The contour interval is 1 Sv.

The average time to complete one circuit varies between 22 years for the first circuit, to 86 years for the seventh circuit (Table 1). This is due to deeper circuits extending more to the east, and the water moving more slowly at depth.

**Cooling and freshening in the Gyre**. To quantify the changes of temperature, salinity and density in the Gyre, Lagrangian divergences of heat, salt and density were computed[9,23] (Fig. 3, see also method section). Lagrangian divergences describe the local changes of heat, salt or density of the water that spirals in the Gyre. The Lagrangian divergences were further differentiated by the instantaneous mixed layer depth in order to quantify if a change is more likely to be caused by air-sea interaction or mixing. Together with the divergences, the Lagrangian barotropic stream function is shown, but only for the water that circulates the Gyre once or more. Since this Lagrangian stream function only includes the northward flowing waters that contribute to the Gyre circulation, it will not be the same as the total Eulerian stream function for the period.

Alongside a general salinity increase, there is a strong cooling in the mixed layer in the Gulf Stream and the northern flank of the Gyre (Fig. 3a, b). This combination results in a large density increase

in the same area that agrees well with the first circuit of the mean trajectory (Fig. 3c and Fig. 1). We therefore suggest that the changes in the first circuit are mainly driven by heat loss to the atmosphere, evaporation and vigorous near-surface mixing. In the same region the NASTMWs is formed[15–18]. NASTMW is formed through convection at the separation of the Gulf Stream and the re-circulation[19] (marked in Fig. 3a). The strong cooling in the northern flank of the Gyre may thus lead to the formation of NASTMW (Fig. 3a, b), and indicates a connection between the formation of NASTMW and the densification in the spiralling Gyre. This inference is further strengthened by noting that the temperature and salinity intervals of NASTMW, often suggested to have a mid-point values around 17 °C, 36.5 g/kg and 26.5 kg/m³[19,24,25], corresponding approximately to our 6th spiral (Fig. 1). Also the depth of NASTMW is in agreement with the depth of the spiral and especially the circuits with temperature and salinities similar to that of NASTMW (Fig. 1). Consequently, the formation of NASTMW and the dynamics of the spiralling nature of these trajectories may be connected and even influence the Gyre circulation and therefore the AMOC. These possible connection motivate further examinations that are beyond the scope of this study.

Below the mixed layer the divergence patterns are very different, for which we identify three distinct regions (marked with numbers

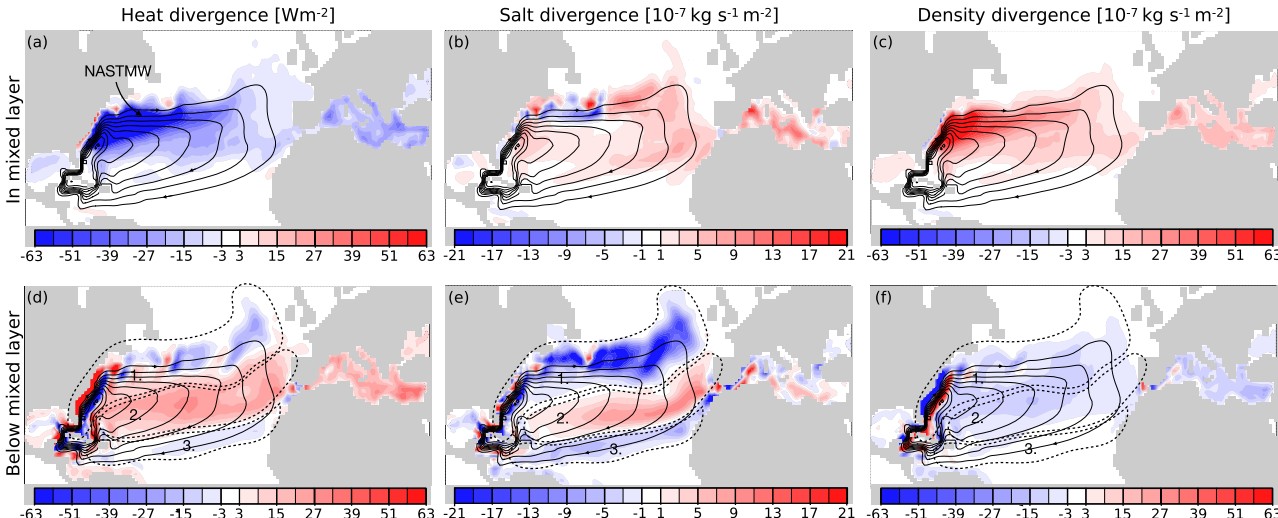

**Fig. 3 The Lagrangian divergence.** Upper panel: The Lagrangian divergences of trajectories that spiral the Gyre in the mixed layer. Superimposed on all figures are the Lagrangian Barotropic stream function computed only for the Gyre circulation. The contour interval of the stream function is 2 Sv. **a–c** The heat, salt and density divergence in the mixed layer computed for the trajectories circuiting the subtropical gyre. Negative values indicate a loss of heat, salt or density. Lower panel: The Lagrangian divergences below the mixed layer depth of the trajectories that spiral in the Gyre. As in the upper panel the Lagrangian Barotropic stream function is superimposed, with contour intervals of 2 Sv. **d–f** The heat, salt and density divergences below the mixed layer for all trajectories circulating the Gyre. Negative values show a loss, whereas positive values indicate a gain. In (**d–f**) three regions are marked as they are discussed in the text as the main regions where changes in heat, salt and density occurs below the mixed layer. The formation region of the North Atlantic Subtropical Mode Water (NASMW) is marked in (**d**).

and contours in Fig. 3d–f)). Region one is the western part of the Gulf Stream and the northern flank of the Gyre, where we observe strong cooling and pronounced freshening (Fig. 3d, e). This effect is also seen in the third and subsequent circuits of the mean trajectory (Fig. 1) and is likely a consequence of interior mixing with denser waters that increase the density and depth of the water parcels. In region one, density surfaces are strongly tilted in combination with strong temperature and salinity gradients, and can thus support lateral mesoscale tracer fluxes[26] below the mixed layer.

The second region is in the middle and eastern parts of the Gyre, here water gains heat and salt and density decreases (Fig. 3d–f). The heat and salt increase is most likely a result of mixing with Mediterranean waters in combination with regional evaporation. The most saline waters of the Atlantic Ocean can be found here[27], even at depth. The third region is in the southern part of the Gyre, where the parcels cool and freshen (Fig. 3d, e). However, these changes are density-compensated, indicating isopycnal mixing as the cause (Fig. 3f). This agrees well with the patterns observed for the second and subsequent circuits of the mean trajectory (Fig. 1).

Since each circuit in the Gyre is a closed system, the amount of heat and salt lost during each circuit can be computed using the related volume transport (Table 1). In the first circuit, a total of $0.15 \cdot 10^6$ kg/s salt is gained by the water, while in the following six circuits a total of $1.37 \cdot 10^6$ kg/s salt is lost, leaving an accumulated salt loss (freshening) of $1.22 \cdot 10^6$ kg/s. Water circulating the Gyre has an accumulated heat loss of 0.24 PW, of which 57% is lost within the first circuit, while the remaining 43% can only be lost by the accumulated effect of multiple circuits. Clearly, a number of Gyre circuits are required for the total water-mass changes to occur and thus escape the Gyre and continue to the northern part of the AMOC. From this we conclude that the accumulated heat loss over multiple circuits around the Gyre has a substantial impact on reducing the northward heat transport in the Atlantic Ocean. These numbers are, however, not comparable with the meridional heat transport computed for one section in the Atlantic Ocean.

**The spiral in a higher resolution model**. We found that the conclusions using a coarse resolution model (1°), also hold when using a high resolution model (1/4°). So far, we used a coarse simulation with a horizontal resolution similar to the CMIP6 models used for climate studies[28], but with a relatively high vertical resolution[29,30]. Unresolved eddies in such coarse resolution models are known to affects the Gulf Stream strength and eddy dynamics[31]. Yet, in high-resolution eddy-resolving models there is a risk of getting a nontrivial heat uptake due to spurious mixing, relative to the coarse resolution models[32].

We here also performed the same analysis with a higher resolution eddy permitting model with a horizontal resolution of 1/4° and 75 depth levels, further on referred to as the HR-case. The CMIP6 resolution run will be referred to as the LR-case. We find that increasing the resolution does not change the above conclusions, but rather strengthen them and show the robustness of the results. Approximately the same amount of transport end up in the Gyre for the HR-case (supplementary Table 1), and the same amount of transport follow each circuit. This is perhaps unsurprising as low-resolution models do well in predicting large-scale and time-mean ocean dynamics. Hence we also find that each circuit show a similar change in heat, salt, density and depth (supplementary Table 1). The spiral in temperature and salinity coordinates are overall very similar, except that the water is slightly more saline in the HR-case, and the circuits are wider (Fig. 4).

It does stand out that in the HR-case it takes about half the time for trajectories to do one circuit compared to the LR-case (supplementary Table 1). This is a result of the more narrow Gulf Stream, transporting the same amount of water in a more narrow region, thus increasing the speed of the water. This, combined with eddies that decrease the size of the circuits, results in a shorter spiralling time (supplementary Fig. 3). These results are thus in line with previous studies releasing trajectories in the Gulf Stream, where the Gulf Stream in low-resolution models is slower than in models with higher resolution[31]. However, even though some differences in the mean trajectory can be seen, the Gyre plays the same role in both the low and high resolution.

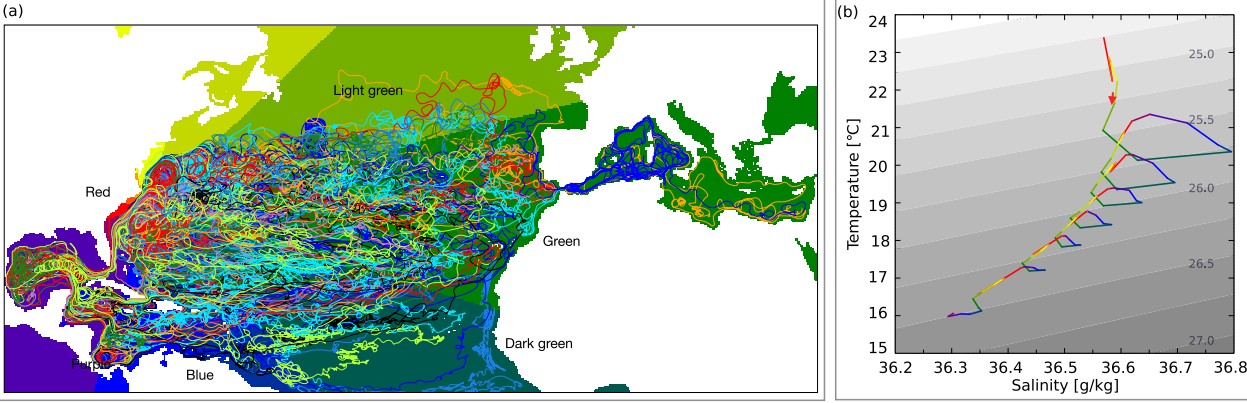

**Fig. 4 The downward spiral in a high resolution model. a** A set of trajectories that circuits the Gyre, simulated using the 1/4° resolution. Only the trajectories circulation in the Gyre is plotted. The colour wheel is used to compute the average trajectory in (**b**) and also in the supplementary material. The naming of the colours are added as they are discussed in the text. **b** The average trajectory in temperature and salinity space for the 1/4° resolution. It is coloured after location following the colour-clock superimposed in (**a**).

## Discussion

We have shown that 70% of the water that flows northwards as part of the AMOC first circulates the Gyre. There it becomes denser due to cooling and freshening and consequently spirals down deeper into the water column. Only when the water is sufficiently dense and deep, it can continue to the northern part of the AMOC. This new insight points to the crucial role of the Gyre for the variability of the AMOC.

We have shown that the whole Gyre is needed both for making the water denser as well as for transporting it back into the Gulf Stream. The main increase in density of the water that circulates in the Gyre takes place in the Gulf Stream. Although significant change in density may take place in the Gyre itself a substantial role of the Gyre could be to simply return the water to the Gulf Stream where it can get sufficiently dense enough to move northwards. This might be seasonally related, where a water parcel arriving to the Gulf Stream during wintertime may undergo larger cooling due to a large heat flux to the atmosphere compared to that arriving during summer. Questions such as this need to be answered in further studies.

Our results suggest that the first circuit in the spiral is influenced by air-sea interactions, while the parcels in the deeper circuits undergo sub-surface mixing. To understand the processes in the latter circuits, a more detailed study concerning mixing processes in the Gyre and thus the Gulf Stream is needed. A next step could be to trace the heat and salt budget terms in the Gyre, to be able to quantify how much of the change in heat and salt that are a result of vertical and lateral diffusion.

For the HR-case the overall results show that the spiralling Gyre is similar, except for some small discrepancies. The heat, salt, density and depth changes are almost identical in both cases. This strengthen the results of the LR-case and also indicate how well the tuning of low-resolution models work, getting heat and salt correct in the models. With a more narrow Gulf Stream and eddies included, the circuits of the Gyre are almost twice as fast in the HR-case.

It is possible that similar circulation patterns could exist in other ocean gyres, since they are driven similarly but influenced by regional and basin scale differences. For example, subduction in the Atlantic Subpolar Gyre that ultimately forms North Atlantic Deep Water, could also be a consequence of waters spiralling downwards. The North Pacific Subtropical Gyre and the Weddell Gyre could possibly generate spiralling patterns. The Southern subtropical gyres span larger areas, and are somewhat

affected by the Antarctic Circumpolar Current, however a spiral similar to that observed in this study is not excluded[33]. Thus, a study on other ocean gyres may be undertaken to further understand the differences and similarities between ocean gyres, and their individual importance for ocean heat and salt.

In a warming climate, vertical stratification in the Gyre is expected to change[5], which could influence the pathways and densification mechanism identified in this study, and therewith affect the AMOC. The spiral shows a clear pattern of salinification in the eastern parts of the Gyre, where evaporation and the overflow of Mediterranean water increases the salinity of the water. Many studies suggest that areas of strong evaporation will become even dryer, which thus would increase the salinity even mores[1]. The already large salinification in each circuit of the gyre would then become even larger, and without any area reducing it, the overall change in salinity in the Gyre would be smaller than now. On the other hand, previous studies[9] suggest a mixing between Subtropical and Subpolar waters in the interface between the gyres, where the Subpolar water increases in salinity as it exchanges salt with the Subtropical waters. The subpolar waters are expected to become fresher due to ice melting, and could therefore play a role in making the water circulating the gyre fresher and thus compensating an increase in salinity from evaporation and mixing with the Mediterranean water.

Previous studies[1] stress the urgency of understanding the changes causing a decrease of the AMOC, but with little or no focus on the role of the Gyre. We emphasise that we also need to understand the circulation patterns and mixing processes in the Gyre, in order to be able to understand the AMOC and the Earth's climate in the future.

## Methods and data

To trace the water masses circulating in the North Atlantic Subtropical Gyre, Lagrangian trajectories were computed with the Lagrangian trajectory model TRACMASS[21,22] using mass flux, temperature and salinity fields from the Earth System Model EC-Earth-Veg version 3.3.1.1[28]. EC-Earth-Veg consists of the atmosphere-land surface module Integrated Forecasting System (IFS) together with a dynamical vegetation model LPJ-GUESS, coupled to the ocean model NEMO3.6[34] (Nucleus for European Modelling of the Ocean) that consists of the ocean model OPA (Océan Parallélisé) and the sea-ice module Louvain-la-Neuve Sea Ice Model (LIM3). This version of NEMO has 75 depth levels and a 1° horizontal resolution, with a tri-polar grid. The resolution implies that the model is not eddy-permitting. Its vertical mixing is set by a Turbulent Kinetic Energy scheme[35], whereas eddy tracer fluxes are parameterised using an adiabatic mixing scheme[36]. The fields are monthly means from the period 1850 to 2014. The mass fluxes used to simulate Lagrangian trajectories in TRACMASS included both the mean velocity

field and the eddy induced velocities[36]. A great advantage with TRACMASS is that the scheme is mass conserving due to the fact that the scheme uses mass fluxes, and not velocities to compute trajectories[22], in the same manner as the NEMO configuration[34]. The conservation of mass makes it possible to compute both Lagrangian stream functions and the Lagrangian divergence.

**Lagrangian simulation**. Trajectories were started at all depths and longitudes at 17°S, where the velocities were northward directed in the Atlantic Ocean, over 1 year to include seasonality. 278,361 trajectories started at 17°S and represented 44.7 Sv (1 Sv = 10 m³ s⁻¹). Each trajectory was associated with a volume transport <2000 m³ s⁻¹. This implies that there were more trajectories started where there was a high volume transport, which thus gives a good Lagrangian resolution of the field[9,9,37–39]. Trajectories were terminated in two ways: (1) if they returned to 17°S or (2) if they reached 58°N. The latter were used in the present study to represent the water that flows northwards as part of the AMOC and the Subtropical Gyre. In total 11 Sv (68,064 trajectories) reached the northern boundary, while 32.7 Sv flowed back to the southern boundary. Out of the trajectories that reached north, 3.3 Sv went directly northwards, whereas the rest circulated the Gyre at least once. Figure 1d shows a selection of the simulated trajectories, and only their circulation within the Gyre. Observations have shown that the measured mean volume transport at 26°N is 18.7 ± 5.6 Sv[40], while at 41°N the transport is lower and measured to 15.5 ± 2.4 Sv[41]. Thus, the northwards flowing waters that we simulated in the present study are a bit lower than observed volume transport, but still in agreement with measurements. These northwards flowing waters are not expected to cover the entire northward flow at 26°N and 41°N as there might be other waters contributing to the total transport. An example would be water originating from the subpolar regions, that reach the tropics but return northwards again, these are not included in the traced water here. In the present study, the total transport northwards excluded those waters that moved northwards but returned back southwards before reaching 58°N. To fully represent the measured mean volume transport at a section, the entire circulation in the Atlantic Ocean has to be traced, including waters originating from the northernmost Atlantic Ocean. Thus, it is expected that the transport of the northwards flowing water is lower than that measured at 26°N, and at 41°N. The trajectory simulation was integrated for 1000 years, which means that the data from 1850 to 2014 were looped. After 1000 years only 0.1 Sv had not reached any boundary and was left in the domain. Finally, trajectories reaching the sea surface and thus evaporating were excluded. This means that there were no sources or sinks of water fluxes through the sea-surface. Due to this, the Lagrangian stream functions will not have any stream lines crossing the sea-surface. A total of 0.9 Sv evaporated.

To test the robustness of the results another Lagrangian simulation was performed, this time with mass flux fields from the EC-Earth3P-HR model[42]. It consists of the atmospheric model IFS and the ocean component NEMO, coupled through the OASIS coupler. The horizontal resolution of the atmospheric model is 40 km, compared to 80 km for the EC-Earth-Veg. The horizontal resolution of the ocean model is increased to 1/4° and 75 vertical levels. The data are stored as monthly means over the period 1950–2014 following the guidelines from HighResMIP[43]. Trajectories were again started at 17°S, and terminated as described above. In total 10 Sv reached the northern boundary at 58°N distributed over 83,337 trajectories.

**Lagrangian stream function**. Since the TRACMASS scheme is mass conserving, Lagrangian stream functions can be computed for the simulated trajectories. A Lagrangian stream function is well defined as long as the mass is conserved. In the present study, all sources and sinks were limited to latitude sections in south and north, and between these sections mass was conserved. This means that we could compute a Lagrangian meridional overturning stream function, using both depth and $\sigma_0$ as coordinates, as follows (Eqs. 1–4):

$$\Psi_{j,k}^{LM} - \Psi_{j,k-1}^{LM} = -\sum_i \sum_n V_{i,j,k,n}^y. \tag{1}$$

Here $V_{i,j,k,n}^y$ is the volume transport crossing a constant latitude $y$. $n$ denotes each trajectory, while $i, j, k$ is longitude, latitude and depth. $k$ can easily be changed to $\sigma_0$ instead, this to get the Lagrangian Meridional Overturning stream function expressed in $\sigma_0$ coordinates. Further, the Lagrangian Barotropic stream function was computed as follows:

$$\Psi_{i,j}^{LB} - \Psi_{i-1,j}^{LB} = \sum_k \sum_n V_{i,j,k,n}^y. \tag{2}$$

Similarly, $V_{i,j,k,n}^y$ is the volume transport crossing a constant latitude $y$.

**The mean trajectory**. To be able to compute the mean trajectory of the spiral, only the part of the trajectories that circulated in the Gyre were selected, (as shown by the trajectories in Fig. 1d). In latitude–longitude space, a colour-clock, making a 12 h revolution, was inserted following Goethe's colour wheel (Fig. 1d or Fig. 5 for a clearer view of the clock). The colour-clock was placed in the middle of the Gyre, so that each colour of the wheel represented a slice of the Gyre (Fig. 5). In this way, the Gyre was separated into 12 slices with different regions. Each trajectory could then be followed in every slice and a mean position of all trajectories was computed for each slice. The first circuit of the mean trajectory was computed by looking at

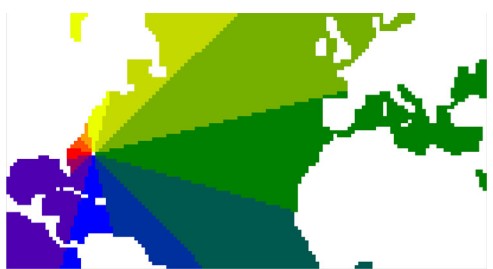

**Fig. 5 The colour-clock.** The colour-clock defined to compute the mean trajectory in Fig. 1. Each colour represents a slice of a clock divided into 12 h. In each slice the mean latitude, longitude, depth, density, temperature and salinity is computed for each circuit that a parcel do.

all trajectories that complete one or more circuits in the Gyre, thereafter the second circuit was computed by looking at all trajectories doing a second loop in the Gyre. The same procedure was followed for each circuit of the Gyre.

For each time-step of the trajectory, its temperature, salinity, density, depth, and position in $x$ and $y$ were recorded and a mean was computed for the colour-clock, using the following equation:

$$\chi_{cl,ci} = \frac{\sum_n \chi_{cl,ci,n}}{\sum_n N_{cl,ci,n}}. \tag{3}$$

Here $\chi_{cl,ci,n}$ represents the parameter that a mean was computed for (e.g. density, depth, latitude, longitude) in the segment of the clock $cl$, the circuit number $ci$, and the traced trajectory $n$. N is the amount of times a point of a trajectory was recorded in the segment of the clock $cl$ for the circuit number $ci$. $cl$ is a number between 1 and 12, which was set by the definition of the clock:

$$cl_{i,j} = \frac{h \cdot \left(180 - \frac{\arctan\left(i - ic, j - jc\right)}{\pi/180}\right)}{360} + \alpha + 1, \tag{4}$$

where $h$ is the number of segments used in the clock, in this case $h = 12$. $ic$ and $jc$ is the central location of the clock according to its $x$ and $y$ coordinates. $\alpha$ is the phase shift of the clock, which was set to 20. These equations were used to compute the 3D spiral in Fig. 1. The mean trajectory was thus a result of several trajectories passing through each segment of the clock for each circuit. To complete the mean trajectory in Fig. 1 only the trajectories that did seven or more loops were used.

**Lagrangian divergence**. The Lagrangian divergence[9,23] provides a detailed view on how much heat, salt or buoyancy a water mass is loosing or gaining in a region. Since TRACMASS is mass conserving it was possible to compute the Lagrangian divergence as follows:

$$\Delta F_{i,j}^T = \sum_k \sum_n V_{i,j,k,n}(T_{i,j,k,n}^{out} - T_{i,j,k,n}^{in}), \tag{5}$$

where $V_{i,j,k,n}$ is the volume transport for trajectory $n$. $T_{i,j,k,n}^{in}$ and $T_{i,j,k,n}^{out}$ is the temperature a trajectory has as it enters and leaves a grid box, respectively. Similarly the Lagrangian divergence of salt and density was computed by changing the temperature ($T$) in Eq. (5) to salinity ($S$) or density ($\sigma_0$). To finally describe the Lagrangian divergence as a heat flux, it was divided by the grid area and multiplied by the heat capacity and density of sea water as follows Eq. (6):

$$H_{i,j}^L = \frac{\rho c_p}{\Delta x_{i,j} \Delta y_{i,j}} \Delta F_{i,j}^T. \tag{6}$$

$H_{i,j}^L$ now describes a heat flux in Wm⁻² that can either be due to heat exchange with the atmosphere through the sea surface or by exchanging heat with another water mass. The same was done for salinity, where Eq. (5) was multiplied with the density of sea water $\rho$, as follows Eq. (7):

$$S_{i,j}^L = \frac{\rho}{\Delta x_{i,j} \Delta y_{i,j}} \Delta F_{i,j}^S. \tag{7}$$

The salt divergence ($S_{i,j}^L$) describes a flux of salt in g s⁻¹ m⁻². To go from g to kg we divided with 10³, so that the salt flux was given in kg s⁻¹ m⁻². This flux in salt can either result from evaporation/precipitation or mixing with another water mass. River runoff can impact the salt distribution of the water masses, but is of no great importance here. In areas where sea ice is present, the salt divergence will change with freezing or melting of ice. Finally, to obtain a density divergence ($D_{i,j}^L$) in kgs⁻¹m⁻² Eq. (5) was used with $\sigma_0$ and divided by the grid area, as follows Eq. (8):

$$D_{i,j}^L = \frac{\rho}{\Delta x_{i,j} \Delta y_{i,j}} \Delta F_{i,j}^D. \tag{8}$$

For each position of the trajectories, the mixed layer depth was saved. This made it possible to trace if a trajectory was in the mixed layer or below in each grid box. With this, the Lagrangian divergences were integrated over two depth-boxes,

one in the mixed layer and one below[23]. The mixed layer used was computed by the ocean model NEMO[34] and defined by a density criterion[44].

## Data availability

The CMIP6 run with EC-Earth-Veg3 and EC-Earth-HR are freely available and can be downloaded from https://esg-dn1.nsc.liu.se/search/cmip6/. The data from the TRACMASS simulation with EC-Earth-Veg3 used to create the results can be downloaded from 10.5281/zenodo.4916433. The data from the TRACMASS simulation with EC-Earth-HR used to create the results can be downloaded from 10.5281/zenodo.6346246.

## Code availability

The Lagrangian trajectory model TRACMASS v7.0 can be downloaded from 10.5281/zenodo.4337926[21]. The codes used to generate the figures in the present study can be downloaded at 10.5281/zenodo.5106459.

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

## Acknowledgements

The authors would like to thank Peter Lundberg for his constructive comments to the paper. The authors would also like to thank Aitor Aldama Campino and Ezra Eisbrenner for comments. The computations and data handling were enabled by resources provided by the Swedish National Infrastructure for Computing (SNIC) at the National Super-computer Centre (NSC) partially funded by the Swedish Research Council through grant agreement no. 2016-07213. S.B and K.D has been financially supported by the Swedish Research Council through grant agreement no. 2019-03574. T.J.McD. gratefully acknowledges Australian Research Council support through grant FL150100090.

## Author contributions

All authors were involved in discussing the results and preparing the final paper. S.B. took the lead on writing the paper with comments from S.G., T.J.McD. and K.D. S.B. defined and ran the Lagrangian simulation in TRACMASS. K.D. computed the mean trajectory. S.B. and K.D. together did the divergence and computed the changes in the Gyre.

## Funding

## Competing interests

The authors declare no competing interests.
