## [Peer Review File · Nature Communications]

Title: The Downward Spiralling Nature of the North Atlantic Subtropical GyreEditorial Note: This manuscript has been previously reviewed at another journal that is not operating a transparent peer review scheme. This document only contains reviewer comments and rebuttal letters for versions considered at *Nature Communications*. Mentions of prior referee reports have been redacted.

REVIEWER COMMENTS

Reviewer #1 (Remarks to the Author):

This is my second review of “The Downward Spiralling Nature of the North Atlantic Subtropical Gyre” by Berglund et al, which has been transferred to Nature Communications from [REDACTED]. The authors have improved the manuscript and addressed the majority of my comments. I believe that it will be well-suited to Nature Communications after a revision.

The authors have now included a connection to subtropical mode water (STMW) in their manuscript, but the details of the connection are still unclear. In their response to my review they show clearly that the model includes subtropical mode water, but this is less clear in the manuscript. This figure would be useful in a supplementary file. In Figure 3d they point to a vague area of STMW formation and imply that it is formed below the mixed layer. Is this where STMW is formed in the model? Are STMW formation areas not typically within the mixed layer and along the northern boundary of the Gulf Stream and its extension? The label may be better suited in panel a. To draw a clear connection between the spiraling trajectories and STMW formation, it would be useful to highlight the area of temperature-salinity space that corresponds to STMW in the model. This could be done in Figures 1c/e, for example, to highlight which loop(s) correspond(s) to STMW on average. It would also be instructive to map the temperature-salinity of all mixed layers in the model run to solidify this connection and clarify how and where STMW are formed in this model.

Minor comments

L11: “AMOC has slowed down during the 20th century” - There is not really scientific consensus on this, direct observations do not show a conclusive slowing. The authors are of course free to state this, however.

L14-18: The writing is quite disjointed here, recommend editing. Could remove the sentence “The connection at depth... in the Subtropical Gyre”.

L43: Recommend changing to “24 to 25” instead.

L85: 22 years seems like a very long time for the first circuit! What are the Gulf Stream and gyre velocities in this model? Can you provide a point of comparison from the literature?

L124: Use of “entrained” is a little strange here. Is the Gulf Stream not part of the gyre? Perhaps a

different word would be more appropriate.

L130: Typo by → be

L156: Add reference to Tamsitt et al. Study about spiralling pathways in the ACC:
<https://www.nature.com/articles/s41467-017-00197-0>

Figure 1 caption: "The naming of the colours are added" → "The names of the colours are included"

L177: $1 \text{ Sv} = 10^6 \text{ m}^3/\text{s}$

L185: Please clarify how the model is in agreement with the observations in this context.

L206: This section is much clearer, thank you. I'm still somewhat confused about whether the trajectories should necessarily connect as different trajectories are used for each circuit. Please clarify this in the text.

Figure 4 seems somewhat superfluous – a different figure clarifying the connection with STMW would be more useful here or in the supplementary information.

Reviewer #2 (Remarks to the Author):

The manuscript "The Downward Spiralling Nature of the North Atlantic Subtropical Gyre" by Berglund et al. describes a Lagrangian study to highlight the role of gyre circulations in the Atlantic Meridional Overturning Circulation (AMOC). I already reviewed the original submission to [REDACTED]. Together with my fellow co-reviewer, we concluded that the subduction process of water spiralling down in the subtropical gyre is a well-known feature. The (original) manuscript contained an illustrative description and visualization, but lacked a deeper physical understanding or wider implications. Our main concern was that the results may not hold at higher resolution.

The revised version, now submitted to Nature Communications, in large parts contains an unchanged manuscript. Apart from some minor points, the step towards a more recent version of EC-Earth and an inclusion of NASTMW, the manuscript again does not address major points of deeper insight or wider implications that would give reason for a publication in a Nature journal. In respect to resolution, the authors "believe that the main conclusions of this paper will hold with increased resolution, considering the fact that these low-resolution models do well in predicting large scale and time mean ocean dynamics. However, an analyses with a higher resolution eddy resolving model, would be beneficial to understand more of the details of the observed trajectories and the mean trajectory." (lines 146-149). I still think that the authors need to address this issue. With such a complicated exchange between subtropical and subpolar gyres that we see from observations and higher resolved models, there is a high chance that the results significantly change.

Apart from the lacking points on deeper insight (physics involved), wider implications (interplay with AMOC) and robustness of the results (resolution issue), there is nothing wrong with this submission. All minor points have been addressed adequately. I leave the final decision to the editors.

Answers to the review of “The Downward Spiralling Nature of the North Atlantic Subtropical Gyre”

The manuscript "The Downward Spiralling Nature of the North Atlantic Subtropical Gyre" by Berglund et al. describes a Lagrangian study to highlight the role of gyre circulations in the Atlantic Meridional Overturning Circulation (AMOC). I already reviewed the original submission to [REDACTED]. Together with my fellow co-reviewer, we concluded that the subduction process of water spiralling down in the subtropical gyre is a well-known feature. The (original) manuscript contained an illustrative description and visualization, but lacked a deeper physical understanding or wider implications. Our main concern was that the results may not hold at higher resolution.

The revised version, now submitted to Nature Communications, in large parts contains an unchanged manuscript. Apart from some minor points, the step towards a more recent version of EC-Earth and an inclusion of NASTMW, the manuscript again does not address major points of deeper insight or wider implications that would give reason for a publication in a Nature journal. In respect to resolution, the authors "believe that the main conclusions of this paper will hold with increased resolution, considering the fact that these low-resolution models do well in predicting large scale and time mean ocean dynamics. However, an analyses with a higher resolution eddy resolving model, would be beneficial to understand more of the details of the observed trajectories and the mean trajectory." (lines 146-149). I still think that the authors need to address this issue. With such a complicated exchange between subtropical and subpolar gyres that we see from observations and higher resolved models, there is a high chance that the results significantly change.

Apart from the lacking points on deeper insight (physics involved), wider implications (interplay with AMOC) and robustness of the results (resolution issue), there is nothing wrong with this submission. All minor points have been addressed adequately. I leave the final decision to the editors.

Answer:

We would like to thank the reviewer for conducting a second round of review of our manuscript and providing us with constructive comments improving the manuscript.

We have now shown that the results from the 1° model holds in a higher resolution model. We have performed the same analysis on the eddy-permitting model EC-Earth-HR, which consist of a ¼° ocean model. Overall, the results from the eddy-permitting model are very similar to those in the 1-degree model.

We have added a section in the manuscript with the spiral in a high-resolution case and added Figure 4. The rest of the figures are provided in the supplementary material for those readers that want to dig deeper in details when comparing the models. We have chosen to do so since our intention here is to justify the results by showing that the spiral exist as moving to higher resolution.

With a higher resolution model, we have now shown that the lack of eddies is not the reason to why water is re-circulating the Gyre instead of moving northwards directly. We have added a bit more discussion around the connection to seasonality. We have also added a bit more of discussion related to mixing with subpolar waters (previous studies has shown a mixing between these waters) and Mediterranean waters (a large part of the spiral in TS show an increase in salinity for the water, connected to Med. Waters). Thus, as the Gyre is a part of the AMOC, AMOCs interplay with subpolar waters and Med. Water gives a wider implication to the study. We think that going beyond this is out of the scope of this paper, but really important for coming studies on the gyre. What happens with this re-circulation in a changing climate, and what will that mean for the future AMOC?

Answers to the review of the manuscript “The Downward Spiralling Nature of the North Atlantic Subtropical Gyre”

This is my second review of “The Downward Spiralling Nature of the North Atlantic Subtropical Gyre” by Berglund et al, which has been transferred to Nature Communications from [REDACTED]. The authors have improved the manuscript and addressed the majority of my comments. I believe that it will be well-suited to Nature Communications after a revision.

The authors have now included a connection to subtropical mode water (STMW) in their manuscript, but the details of the connection are still unclear. In their response to my review they show clearly that the model includes subtropical mode water, but this is less clear in the manuscript. This figure would be useful in a supplementary file. In Figure 3d they point to a vague area of STMW formation and imply that it is formed below the mixed layer. Is this where STMW is formed in the model? Are STMW formation areas not typically within the mixed layer and along the northern boundary of the Gulf Stream and its extension? The label may be better suited in panel a. To draw a clear connection between the spiraling trajectories and STMW formation, it would be useful to highlight the area of temperature-salinity space that corresponds to STMW in the model. This could be done in Figures 1c/e, for example, to highlight which loop(s) correspond(s) to STMW on average. It would also be instructive to map the temperature-salinity of all mixed layers in the model run to solidify this connection and clarify how and where STMW are formed in this model.

Answer:

We would like to sincerely thank reviewer 1 for conducting a second review of our manuscript, providing good comments to improve our manuscript.

The review touches mainly upon the connection to STMW. We have added a bit more about the STMW and the connection to the spiral we have shown. Mainly connecting to formation areas and to temperature and salinities of STMW. We believe that this connection is enough in our manuscript, as the scope is not to trace STMW but rather show the mechanism of the spiraling Gyre. However, we agree with the reviewer that this connection is important, and that’s why we have added a bit more about the connection. We also believe that future studies need to target this connection even more.

Minor comments

L11: “AMOC has slowed down during the 20th century” - There is not really scientific consensus on this, direct observations do not show a conclusive slowing. The authors are of course free to state this, however.

Answer: We have rephrased this, as the consensus points to a slow down, but it is unclear how strong it is.

L14-18: The writing is quite disjointed here, recommend editing. Could remove the sentence “The connection at depth... in the Subtropical Gyre”.

Answer: We have edited this section so the connection between the sentences is better.

L43: Recommend changing to “24 to 25” instead.

Answer: We have changed to “24 to 25”.

L85: 22 years seems like a very long time for the first circuit! What are the Gulf Stream and gyre velocities in this model? Can you provide a point of comparison from the literature?

Answer: With the higher resolution we get half the speed, suggesting the low resolution to be much slower than the higher. Probably due to the wide Gulf Stream with similar volume transport, thus lower velocities. This can for example be seen in Nilsson et al 2013 (which we have added as reference as we discuss this for the new high-resolution run).

L124: Use of “entrained” is a little strange here. Is the Gulf Stream not part of the gyre? Perhaps a different word would be more appropriate.

Answer: We have changed “entrained”, and instead written “...AMOC first circulates the Gyre.”

L130: Typo by → be

Answer: We have change by to be.

L156: Add reference to Tamsitt et al. Study about spiralling pathways in the

ACC: <https://www.nature.com/articles/s41467-017-00197-0>

Answer: We have added the reference.

Figure 1 caption: “The naming of the colours are added” → “The names of the colours are included”

Answer: We rephrased the sentence as suggested.

L177: $1 \text{ Sv} = 10^6 \text{ m}^3/\text{s}$

Answer: We have fixed this typo.

L185: Please clarify how the model is in agreement with the observations in this context.

Answer: We have clarified that the results are a bit lower, but still close to observations. We have also clarified that not all water is included here, since there might be water originating from somewhere else that contribute to a Eulerian northward flow.

L206: This section is much clearer, thank you. I’m still somewhat confused about whether the trajectories should necessarily connect as different trajectories are used for each circuit. Please clarify this in the text.

Answer: We have tried to clarify this more. For the mean trajectory, only trajectories that completed seven or more loops were included. Thus, only the same trajectories did the entire spiral. However, for computing the total change of heat, salt and density (used in the Table and for the divergence) all trajectories that completed each loop where included. E.g., for the 3rd circuit both those that did exactly 3 circuits together with those doing 3+ circuits where used.

Figure 4 seems somewhat superfluous – a different figure clarifying the connection with STMW would be more useful here or in the supplementary information.

Answer: We have decided to keep figure 4 (now figure 5) as it shows the clock clearer than in Figure 1, and thus seems necessary for understanding the method. We have decided to not add any figure on STMW, but added more on the topic in the manuscript.

REVIEWERS' COMMENTS

Reviewer #2 (Remarks to the Author):

Thank you for revising manuscript. The current version now provides a solid and robust submission, and I would recommend the publication in Nature Comm.

One detail I still found lacking: Neither the model description (Methods and Data) nor supplementary material do contain a model description (or even reference) of the new $1/4^\circ$ (HR) experiment. For completeness, this is requested.

Respos to reviewer

We would like to thank the reviewer for going through our manuscript a final time. We are below providing an answer to the points addressed by the reviewer.

1. One detail I still found lacking: Neither the model description (Methods and Data) nor supplementary material do contain a model description (or even reference) of the new $1/4^\circ$ (HR) experiment. For completeness, this is requested.

- We have added a reference and some more details in the methods-section about the HR experience and the model.